# Severe COVID-19 pneumonia: Perfusion analysis in correlation with pulmonary embolism and vessel enlargement using dual-energy CT data

**Florian Poschenrieder** [1]*, **Stefanie Meiler**[1], **Matthias Lubnow**[2], **Florian Zeman**[3], **Janine Rennert**[1], **Gregor Scharf**[1], **Jan Schaible**[1], **Christian Stroszczynski**[1], **Michael Pfeifer**[2,4], **Okka W. Hamer**[1,5]

1 Department of Radiology, Regensburg University Medical Center, Regensburg, Germany, 2 Department of Internal Medicine II, Regensburg University Medical Center, Regensburg, Germany, 3 Center for Clinical Studies, Regensburg University Medical Center, Regensburg, Germany, 4 Department of Pneumology, Donaustauf Hospital, Donaustauf, Germany, 5 Department of Radiology, Donaustauf Hospital, Donaustauf, Germany

* florian.poschenrieder@ukr.de

**Data Availability Statement:** All relevant data are within the manuscript and its Supporting Information files.

## Abstract

### Background

Gas exchange in COVID-19 pneumonia is impaired and vessel obstruction has been suspected to cause ventilation-perfusion mismatch. Dual-energy CT (DECT) can depict pulmonary perfusion by regional assessment of iodine uptake.

### Objective

The purpose of this study was the analysis of pulmonary perfusion using dual-energy CT in a cohort of 27 consecutive patients with severe COVID-19 pneumonia.

### Method

We retrospectively analyzed pulmonary perfusion with DECT in 27 consecutive patients (mean age 57 years, range 21–73; 19 men and 8 women) with severe COVID-19 pneumonia. Iodine uptake (IU) in regions-of-interest placed into normally aerated lung, ground-glass opacifications (GGO) and consolidations was measured using a dedicated postprocessing software. Vessel enlargement (VE) within opacifications and presence of pulmonary embolism (PE) was assessed by subjective analysis. Linear mixed models were used for statistical analyses.

### Results

Compared to normally aerated lung 106/151 (70.2%) opacifications without upstream PE demonstrated an increased IU, 9/151 (6.0%) an equal IU and 36/151 (23.8%) a decreased IU. The estimated mean iodine uptake (EMIU) in opacifications without upstream PE (GGO 1.77 mg/mL; 95%-CI: 1.52–2.02; p = 0.011, consolidations 1.82 mg/mL; 95%-CI: 1.56–2.08, p = 0.006) was significantly higher compared to normal lung (1.22 mg/mL; 95%-CI: 0.95–1.49).

**Funding:** The authors received no specific funding for this work.

**Competing interests:** The authors have declared that no competing interests exist.

**Abbreviations:** COVID-19, coronavirus disease 2019; EMIU, estimated mean iodine uptake (mg/mL); GGO, ground-glass opacification; IU, iodine uptake (mg/mL); PBV, pulmonary perfused blood volume; PE, pulmonary embolism; ROI, region of interest; RT-PCR, reverse transcription polymerase chain reaction; SARS-CoV-2, severe acute respiratory syndrome coronavirus 2; VE, vessel enlargement.

In case of upstream PE, EMIU of opacifications (combined GGO and consolidations) was significantly decreased compared to normal lung (0.52 mg/mL; 95%-CI: -0.07–1.12; p = 0.043). The presence of VE in opacifications correlated significantly with iodine uptake (p<0.001).

## Conclusions

DECT revealed the opacifications in a subset of patients with severe COVID-19 pneumonia to be perfused non-uniformly with some being hypo- and others being hyperperfused. Mean iodine uptake in opacifications (both ground-glass and consolidation) was higher compared to normally aerated lung except for areas with upstream pulmonary embolism. **Vessel enlargement correlated with iodine uptake**: In summary, in a cohort of 27 consecutive patients with severe COVID-19 pneumonia, dual-energy CT demonstrated a wide range of iodine uptake in pulmonary ground-glass opacifications and consolidations as a surrogate marker for hypo- and hyperperfusion compared to normally aerated lung. Applying DECT to determine which pathophysiology is predominant might help to tailor therapy to the individual patient´s needs.

## Introduction

Oxygenation of patients suffering from COVID-19 pneumonia is reported to be difficult due to a ventilation-perfusion mismatch [1–4]. Over the course of the SARS-CoV-2 pandemic we had to learn that COVID-19 is associated with a high risk of thrombotic and embolic events notably in the second, hyperinflammatory stage of disease [5–12]. Autopsy studies showed endothelial injury and widespread thrombosis with microangiopathy [13, 14]. Thus, ventilation-perfusion mismatch is thought to be mainly due to dead space ventilation. Therefore, aggressive anticoagulation is meanwhile accepted part of therapy [15]. Somewhat contradictory to the outlined pathophysiology is the fact that vessel enlargement within pulmonary opacifications is a frequent finding on CT indicating rather hyperaemia than vessel occlusion [16–18]. Ventilation-perfusion mismatch might therefore also be caused by shunting [1, 3].

Dual-energy computed tomography (DECT) pulmonary angiography is an established tool to assess the regional distribution of pulmonary perfusion [19–21]. Several groups applied DECT to analyze pulmonary perfusion patterns in COVID-19 pneumonia [22–26]. The results are inhomogeneous. Si-Mohamed et al. found lobes with predominant GGO (early phase of disease) to be hyperperfused and lobes with predominant consolidation (late phase of disease) to be hypoperfused [22]. On the contrary, Ridge saw a decreasing amount of perfusion defects after day 14 [23]. Grillet et al. reported hyperperfusion of both ground glass opacities (GGO) and consolidation [24]. Lang et al. reported about a mixture of hypo- and hyperperfused opacifications [26]. The latter two groups did not provide any data about the phase of disease. Different disease phases, extremely inhomogeneous patient groups with different portions of mildly and severely diseased patients, different modes of analysis (subjective versus objective) and missing correlation with upstream pulmonary embolism might be the reasons for the conflicting results.

The purpose of this study was the objective analysis of pulmonary perfusion using DECT in a cohort of 27 consecutive patients with severe COVID-19 pneumonia including correlation with vessel enlargement and pulmonary embolism.

## Materials and methods

From April 1st, 2020 all CTs of the chest in patients with severe COVID-19 pneumonia (dyspnea, respiratory frequency $\geq$ 30/min, blood oxygen saturation $\leq$ 93%, partial pressure of arterial oxygen to fraction of inspired oxygen ratio < 300, and / or lung infiltrates > 50% within 24 to 48 hours) [27] were performed contrast-enhanced using dual-energy mode. These prospectively acquired data were retrospectively evaluated. The study was approved by the institutional ethics committee (ethics committee of the Regensburg University Medical Center, IRB nr. 20-1784-104). Written informed consent was waived. All procedures performed in studies involving human participants were in accordance with the ethical standards of the institutional and / or national research committee and with the 1964 Helsinki declaration and its later amendments or comparable ethical standards.

### Study population

The inclusion criteria were consecutive adult patients ($\geq$ 18 years old) with RT-PCR-proven SARS-CoV-2 infection and COVID-19 pneumonia who had a contrast-enhanced DECT of the chest performed between April 1st and May15th 2020 at our university tertiary care medical center. Image quality of all examinations was rated before analysis. Exclusion criteria were non-diagnostic CTs due to technical artifacts. In case of more than one CT per patient, the first CT only was analyzed. Patients were identified by means of a full-text database query of all CT-scans performed between April 1st and May 15th 2020 using the term "*COVID*" OR *SARS* in the Radiological Information System (RIS, Nexus.med RIS, Version 8.42, Nexus, Villingen-Schwenningen, Germany) and by searching the electronic patients records for SARS-CoV-2 RT-PCR results. Patient characteristics (age, gender, date of symptom onset, referring ward and ventilation and / or ECMO status) and anticoagulation and corticosteroid therapy status at the time of the dual-energy CT were extracted from electronic patient records. CT imaging parameters including contrast medium injection modus were extracted from the DICOM protocols of the scans.

### CT protocol

All patients underwent contrast-enhanced DECT of the chest with or without additional scans of the head and / or abdomen. All DECT examinations were performed in supine position during end-inspiratory hold using a second generation dual-source CT scanner (Siemens Somatom Definition Flash; Siemens Healthcare, Erlangen, Germany). The contrast medium was iohexol (Accupaque 350, GE Healthcare Buchler, Braunschweig, Germany) administered at a flow rate of 3–5 mL/s. 70 mL were injected for examinations of the chest only. 100 mL were injected for chest examinations in combination with other body regions. The scan was triggered with a bolus-tracking technique. For this, a region of interest (ROI) was placed into the main pulmonary artery. The scan started with a scan delay of 5 seconds after reaching a threshold of 100 Hounsfield Units (HU). All DECT were performed with 100 kVp (tube A) and 140 kVp with tin filter (tube B) with a quality reference mA setting of 150 mA (for tube A) using the combined angular and longitudinal automatic tube current modulation technique (CARE Dose 4D, Siemens Healthcare, Erlangen, Germany).

### Image postprocessing

From each DECT data set perfusion images were calculated using a dedicated dual-energy postprocessing software for assessment of pulmonary perfused blood volume (PBV) (syngo. CT Dual Energy, version VB30A, Siemens Healthcare, Erlangen, Germany). The preset

parameters for tissue recognition were minimum -960 HU and maximum -600 HU. These values are the default setting for evaluation of normally aerated lung and do not allow for analysis of opacified lung. Thus, the maximum was manually set to +150 HU to also include ground glass opacities and consolidation into the analysis. Normalization was performed by placing a dedicated ROI into the main pulmonary artery. Perfusion images were reconstructed as maximum intensity projections (MIP) in three planes (axial, coronal, sagittal) with a slice thickness of 5 mm and an increment of 5 mm. For colour-coding the look-up table (LUT) overlay „Spectrum 10"was applied. Distribution of colours were set in such a way that another colour was displayed for an enhancement increase of 8 HU each. Moreover, multiplanar reconstructions (MPR) applying lung (B60f) and soft-tissue (I31f/2) kernel were generated from weighted average images (60% from the 140 kVp and 40% from the 100 kVp data). MPRs were reconstructed in three planes (axial, coronal, sagittal) with a slice thickness of 1 mm and an increment of 0.75 mm. The images were sent to a picture archiving and communication system (PACS, Syngo Imaging, Siemens Healthcare, Erlangen, Germany).

## Image analysis

In a first step, overall image quality considering motion and beam hardening artifacts, contrast-noise ratio and opacification of pulmonary arteries of all dual-energy CT examinations meeting the inclusion criteria was subjectively assessed by two readers in consensus: one board-certified general radiologist with 12 years of experience (JR) and one subspeciality thoracic radiologist with 15 years of experience (FP), using a four-point scale: 1, excellent, no artifacts; 2, good, mild impairment of image quality without impairment of evaluability; 3, satisfactory, moderate impairment of image quality without impairment of evaluability; and 4, inadequate, severe artifacts, not evaluable for pulmonary perfusion analysis. Dual-energy image analysis was performed in consensus by a subspecialty thoracic radiologist with 15 years of experience (FP) and a board-certified general radiologist with 12 years of experience (JR). Both readers were blinded to clinical patient status. Colour-coded pulmonary perfused blood volume (PBV) images were displayed side by side with MPR in lung and soft-tissue window. The colour-coded PBV maps were displayed as axial maximum intensity projections at 5 millimeters reconstruction slice thickness. MPRs were analyzed regarding the presence and distribution of GGO and consolidation. The ROIs for consolidation and GGO were drawn as large as possible. Mixed areas were intentionally excluded in order to collect data purely representing consolidation and GGO, respectively. Pulmonary embolism (PE) was recorded. The diameter of vessels within opacifications was evaluated. Vessel enlargement was defined as clearly larger vessel diameter compared to vessels of the same generation in normally aerated lung tissue. Also, pulmonary embolism (PE) upstream of opacification was recorded.

Iodine uptake (mg/mL lung tissue) was measured 1. by placing a ROI in normally aerated lung, 2. by placing a ROI in each of up to three separate areas of GGO and 3. by placing a ROI in each of up to three separate areas of consolidation. The decision which areas of GGO and consolidation were analyzed was made according to the colour-coded PBV map. Opacifications with least, medium and most intense enhancement were selected in order to record the entire range of perfusion of GGO and consolidation, respectively.

## Statistical analysis

Categorical data are presented as absolute and relative frequencies, while continuous data are presented using mean, standard deviation (SD) and range (min-max). Differences in the iodine uptake between normally aerated lung and opacified lung (separated into GGO and consolidation), between vessel enlargement and no vessel enlargement, and between iodine

uptake downstream to pulmonary embolism and no pulmonary embolism, were analyzed by using linear mixed models. To account for repeated measurements of patients within the GGO and consolidation areas, patients nested within the respective area were added as a random factor to the models. Estimated means and corresponding 95%-confidence intervals are presented as effect estimates. A p-value <0.05 was considered statistically significant for all analyses. All analyses were performed by using SAS (Version 9.4, The SAS institute, Cary, NC) and the procedure *proc mixed*.

# Results

## Patient population

29 patients were eligible. The CTs of 2 patients were excluded due to severe technical artifacts (classified as score of 4 in overall image quality assessment, see below). Thus 27 patients (19 men (70%) and 8 women (30%)) were included in the study. Mean age was 57 years (SD 13 years, range 21–73 years). The mean interval from symptom onset to the date of the DECT was 23.4 days (SD 12.8 days; range: 7–62 days). At the time of DECT, 26 patients (96%) were treated on the intensive care unit, and one patient (4%) was treated on the general ward. 24 patients (89%) were mechanically ventilated, and 9 patients (33.3%) were on extracorporeal membrane oxygenation. The clinical indication for DECT was critical worsening of symptoms in 25 patients (92.6%) and suspicion of pulmonary embolism in 5 patients (18.5%). At the time of DECT, all 27 patients had at least prophylactic anticoagulation therapy, with 16 patients (59.3%) receiving full-dose therapeutic anticoagulation and 2 patients (7.4%) receiving half-therapeutic dose anticoagulation. At the time of DECT, 8 patients (29.6%) received corticosteroid therapy. 20 DECT examinations were combined scans of several body regions (head, abdomen). 7 scans were chest examinations only. Patient demographics are summarized in Table 1.

## Image analysis

Overall image quality of the evaluated dual-energy CTs was excellent (score of 1) in 7 cases (24.1%), good (score of 2) in 16 cases (55.2%), satisfactory (score of 3) in 4 cases (13.8%) and inadequate (score of 4) in 2 cases (6.9%). The two cases with score 4 were excluded from analysis. The estimated mean iodine uptake of normally aerated lung was 1.22 mg/mL (95%-CI 0.95, 1.49). Three different areas of GGO could be analyzed in all 27 scans, resulting in 81 measurements. As for consolidation, three measurements could be made in 25 scans and two measurements in 2 scans, resulting in 79 measurements (Table 2). Fig 1 illustrates the range of iodine uptake of both GGO and consolidation displayed as difference to normally aerated lung for each scan.

**Pulmonary perfused blood volume (PBV) without upstream pulmonary embolism.** There was a considerable inhomogeneity of iodine density in opacifications even when focusing on opacifications without upstream PE (Table 3). The estimated mean iodine uptake for GGO and consolidation without evidence of upstream PE was higher than that in normally

**Table 1. Patient demographics.**

| Patient demographics | |
|---|---|
| Age (years), mean (SD, range) | 57 (13, 21–73) |
| Gender | 19 m (70%), 8 w (30%) |
| Days from symptom onset to date of DECT, mean (SD, range) | 23.4 (12.8, 7–62) |
| Patients on intensive care unit | 26 (96%) |
| Patients mechanically ventilated | 24 (89%) |
| Patients on general ward | 1 (4%) |

**Table 2. Overview of measurements of iodine uptake and CT findings.**

| Patient# | Age (years) | Gender | Interval from symptom onset to CT (days) | Iodine uptake (mg/mL) | | | | | | | | | | | | | | | | | | |
|---|---|---|---|---|---|---|---|---|---|---|---|---|---|---|---|---|---|---|---|---|---|---|
| | | | | Normal lung | GGO #1 | VE GGO #1 | PE GGO #1 | GGO #2 | VE GGO #2 | PE GGO #2 | GGO #3 | VE GGO #3 | PE GGO #3 | CO #1 | VE CO #1 | PE CO #1 | CO #2 | VE CO #2 | PE CO #2 | CO #3 | VE CO #3 | PE CO #3 |
| 1 | 68 | m | 16 | 0.3 | 0.5 | + | - | 1 | + | - | 0.8 | + | - | 1.5 | + | - | 0.2 | - | + | 0.7 | - | - |
| 2 | 65 | m | 23 | 0.3 | 1.5 | + | - | 0.4 | - | - | 1.5 | + | - | 1.8 | + | - | 1.4 | + | - | 0.1 | - | - |
| 3 | 68 | w | 24 | 2.1 | 2.7 | + | - | 2.7 | + | - | 2.5 | + | - | 1.9 | - | - | 1.8 | - | - | 2.1 | + | - |
| 4 | 60 | m | 23 | 1.2 | 1 | - | - | 0.4 | - | - | 0.7 | - | - | 1.5 | + | - | 0.3 | - | - | 0.6 | - | - |
| 5 | 68 | w | 16 | 1.6 | 1.3 | - | - | 1.1 | - | - | 2.6 | + | - | 1.4 | - | - | 2.2 | + | - | 3.3 | + | - |
| 6 | 63 | m | 17 | 1.3 | 1.4 | + | + | 1.9 | + | + | 2.6 | + | - | 0.3 | + | + | 0.3 | - | + | 0.1 | - | + |
| 7 | 73 | w | 7 | 0.9 | 2.5 | + | - | 1.7 | + | - | 2.3 | + | - | 0.8 | + | + | 1.8 | + | - | 2.2 | + | - |
| 8 | 64 | w | 15 | 1.6 | 1.6 | + | - | 1.7 | + | - | 2.1 | + | - | 0.9 | - | - | 1.8 | + | - | 1.9 | + | - |
| 9 | 36 | m | 22 | 1.5 | 2.4 | + | - | 4.2 | + | - | 2.6 | + | - | 2.1 | - | - | 4.3 | - | - | 0.2 | - | + |
| 10 | 68 | m | 29 | 0.1 | 2.4 | + | - | 3.1 | + | - | 1.4 | - | - | 0.1 | - | - | 2.2 | - | - | 2.3 | - | - |
| 11 | 39 | m | 10 | 1.5 | 2.6 | + | - | 1.6 | + | - | 2.4 | + | - | 1.2 | + | - | 2 | - | - | 3 | + | - |
| 12 | 57 | m | 8 | 1.7 | 1.5 | + | - | 1.4 | + | - | 1.8 | + | - | 1.3 | + | - | 1.2 | + | - | X | X | X |
| 13 | 31 | m | 8 | 1 | 0.7 | - | - | 2 | + | - | 1.7 | + | - | 1.2 | - | - | 1.5 | + | - | 1.5 | - | - |
| 14 | 51 | m | 62 | 1.5 | 1.1 | - | - | 1.5 | + | - | 1.6 | + | - | 2.9 | + | - | 0.8 | - | - | 0.7 | - | - |
| 15 | 64 | w | 52 | 1.2 | 1.6 | + | - | 2.1 | + | - | 3 | + | - | 2.3 | + | - | 2.6 | + | - | 1.8 | - | - |
| 16 | 62 | m | 25 | 0.7 | 0.8 | + | - | 0.8 | + | - | 1.3 | + | - | 0.6 | - | - | 1.7 | + | - | 0.5 | - | - |
| 17 | 73 | m | 29 | 0.7 | 1.2 | - | - | 1.6 | + | - | 1 | - | - | 0.8 | - | - | 1.3 | + | - | 1.6 | + | - |
| 18 | 69 | m | 13 | 1.1 | 0.8 | - | - | 0.5 | - | - | 1.1 | + | - | 1.4 | - | - | 2 | + | - | 0.4 | - | + |
| 19 | 55 | m | 19 | 2.2 | 2.4 | + | - | 1 | + | - | 2.2 | + | - | 1.6 | - | - | 2.4 | + | - | 2.5 | + | - |
| 20 | 55 | m | 25 | 0.9 | 0.9 | - | - | 1.8 | + | - | 2.3 | + | - | 2.5 | + | - | 1.7 | + | - | 0.9 | - | - |
| 21 | 44 | w | 15 | 1.1 | 2.3 | + | - | 1.6 | + | - | 2.2 | + | - | 3 | + | - | 2.8 | + | - | 3.6 | + | - |
| 22 | 57 | m | 40 | 1.9 | 2.8 | + | - | 2.2 | - | - | 1.7 | - | - | 4 | + | - | 1.7 | - | - | 1.4 | - | - |
| 23 | 55 | m | 36 | 1.3 | 0.7 | - | - | 0.3 | - | - | 1.6 | + | - | 1 | - | - | 1.6 | + | - | 1 | + | - |
| 24 | 51 | m | 22 | 1.8 | 1.9 | - | - | 3.8 | + | - | 3.4 | + | - | 3.4 | + | - | 3.2 | + | - | 3.4 | + | - |
| 25 | 67 | m | 31 | 0.4 | 1.3 | + | - | 2 | + | - | 1.5 | + | - | 1.5 | + | - | 1.7 | + | - | 0.7 | - | - |
| 26 | 21 | w | 19 | 2.2 | 2.9 | + | - | 3.8 | + | - | 2.1 | - | - | 5.1 | + | - | 0.2 | - | - | X | X | X |
| 27 | 51 | m | 26 | 0.9 | 1 | - | - | 1 | + | - | 1.5 | + | - | 0.9 | - | - | 2.4 | + | - | 2 | + | - |

f: female.

m: male.

ROI: Region of interest, measurement area.

#: Number (of patient, or of measurement area, respectively).

GGO: Ground glass opacification.

VE: Vessel Enlargement proximal to ROI.

PE: Pulmonary Embolism proximal to ROI.

CO: Consolidation.

+: positive finding.

-: negative finding.

x: missing value.

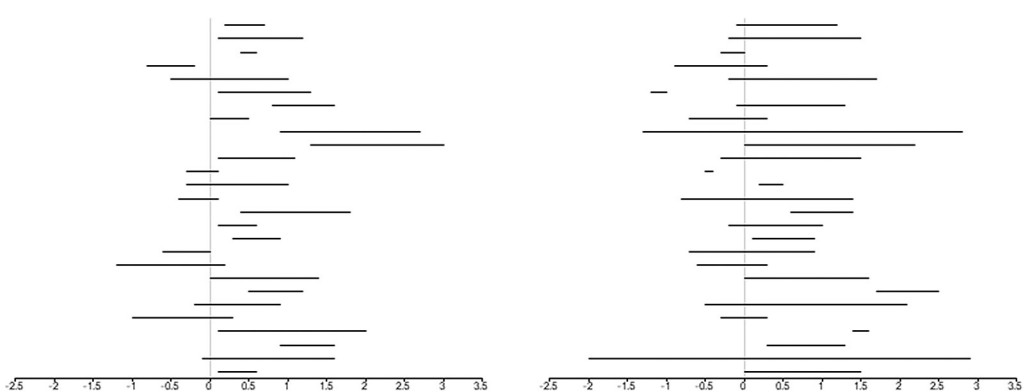

iodine uptake: ground glass vs. normally aerated lung iodine uptake: consolidation vs. normally aerated lung

**Fig 1. a.** Difference of maximum and minimum of iodine uptake (mg/mL; x-axis) in ground-glass opacification compared to normally aerated lung. Each horizontal line represents one patient (y-axis; order of patients from top to bottom as in Table 2). **b.** Difference of maximum and minimum of iodine uptake (mg/mL; x-axis) in consolidation compared to normally aerated lung. Each horizontal line represents one patient (y-axis; order of patients from top to bottom as in Table 2).

aerated lung: GGO 1.77 mg/mL (95%-CI: 1.52–2.02), consolidation: 1.82 mg/mL (95%-CI:1.56–2.08) (shown in Figs 2–5). The difference of the estimated mean iodine uptake values of normally aerated and opacified lung was significant for both GGO (p = 0.011) and consolidation (p = 0.006). In contrast, no significant difference was seen regarding the iodine uptake of ground glass and consolidation (p = 0.791).

**PBV <u>without</u> upstream pulmonary embolism and vessel enlargement.** Vessel enlargement (VE) was present in 57/79 (72.2%) of GGO areas and in 41/72 (56.9%) of consolidated areas without upstream PE. When VE was present without upstream PE, GGO was hyperperfused in 50/57 (87.7%) and equally or hypoperfused compared to normally aerated lung in 4/57 (7%) and 3/57 (5.3%), respectively. When VE was present, consolidation was hyperperfused in 36/41 (87.8%) and equally or hypoperfused compared to normally aerated lung in 1/41 (2.4%) and 4/41 (9.8%), respectively. Estimated mean iodine uptake of areas of GGO demonstrating VE was 2.01 mg/mL (95%-CI: 1.74–2.28) and estimated mean iodine uptake of areas of consolidation with accompanying VE was 2.28 mg/mL (95%-CI: 1.98–2.57). Without accompanying VE, the estimated mean iodine uptake of areas of GGO and of consolidation was 1.15 mg/mL (95%-CI: 0.79–1.50) and 1.21 mg/mL (95%-CI: 0.9–1.53). Statistical analysis showed that the estimated mean iodine uptake of opacified lung areas (GGO and consolidation) with VE was significantly higher as compared to estimated mean iodine uptake of areas

**Table 3. Iodine uptake of opacifications compared to normally aerated lung.**

| Iodine uptake* | Ground glass | | | Consolidation | | |
|---|---|---|---|---|---|---|
| | total | Vessel enlargement | Pulmonary embolism | total | Vessel enlargement | Pulmonary embolism |
| increased | 60 | 52 | 2 | 48 | 36 | 0 |
| equally | 5 | 4 | 0 | 4 | 1 | 0 |
| decreased | 16 | 3 | 0 | 27 | 6 | 7 |
| total | 81 | 59 | 2 | 79 | 43 | 7 |

* Relative to iodine uptake in normally aerated lung.

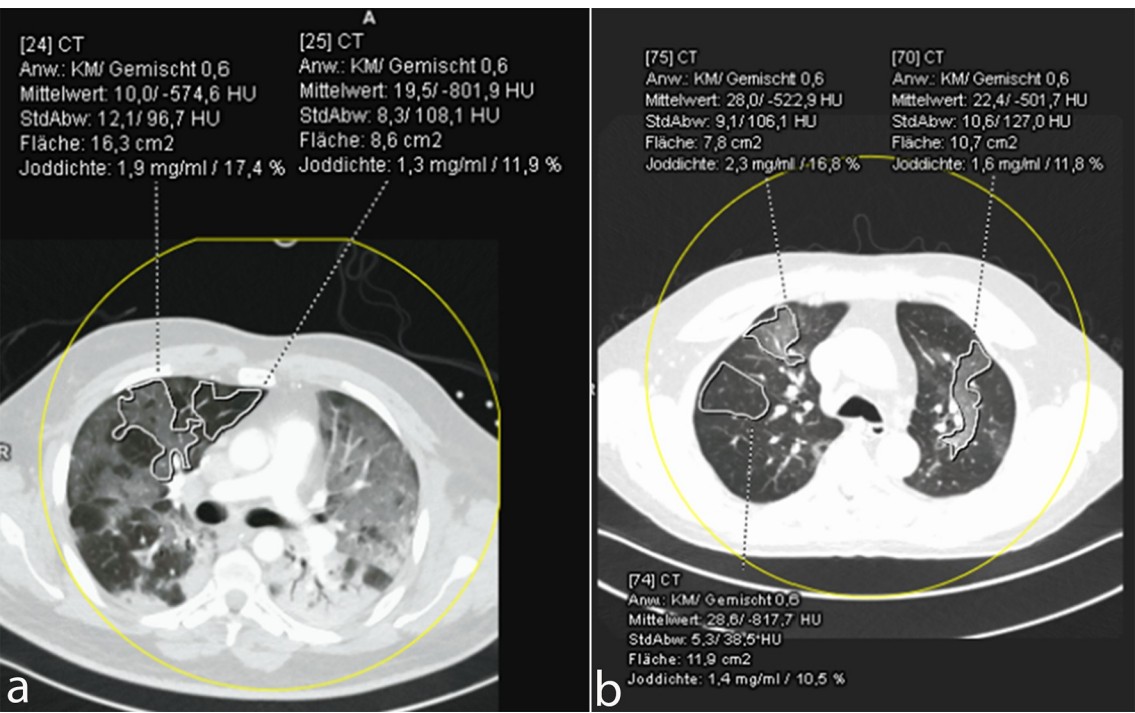

**Fig 2.** Iodine uptake in normally aerated lung and ground-glass opacifications (two different patients, a) and b). Ground-glass opacifications demonstrated a higher iodine density as compared to normally aerated lung.

of GGO and consolidation with normal vessel width (p<0.001) (shown in Fig 6). The estimated mean iodine uptake of opacified lung areas (GGO and consolidation) without VE did not differ from that of normally aerated lung.

**PBV <u>with</u> upstream pulmonary embolism.** PE upstream to the measurement ROI was present in 2/81 (2.5%) of GGO and in 7/79 (8.9%) of consolidated areas. The estimated mean iodine uptake of areas with GGO and consolidation (combined) without and with evidence of upstream PE was 1.79 mg/mL (95%-CI: 1.62–1.97) and 0.52 mg/mL (95%-CI: -0.07–1.12), respectively. The difference was significant (p<0.001).

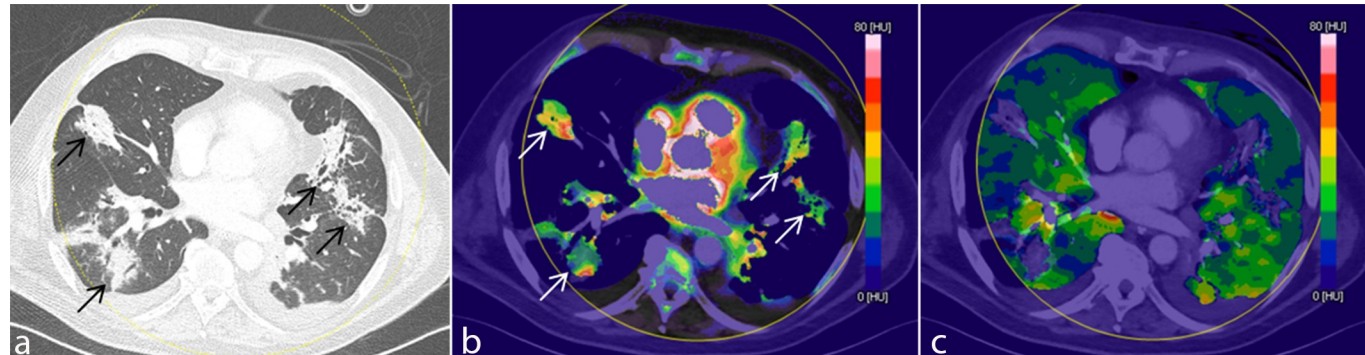

**Fig 3. Iodine uptake in normally aerated lung and consolidation.** Three reconstructions of the identical slice position, a. MPR in lung window, b. colour-coded PBV MIP with thresholds for tissue recognition set to minimum -100 HU and maximum + 150 HU for visualization of iodine density in consolidation and c. colour-coded PBV MIP with thresholds for tissue recognition set to minimum -960 HU and maximum -600 HU for visualisation of iodine density in normally aerated lung. Parameter settings were separated for this illustration only in order to enable a clear depiction of perfusion differences by optimizing the range of the look-up table to tissue density. Consolidations (arrows) demonstrated a higher iodine density as compared to normally aerated lung (see colour-coding scale).

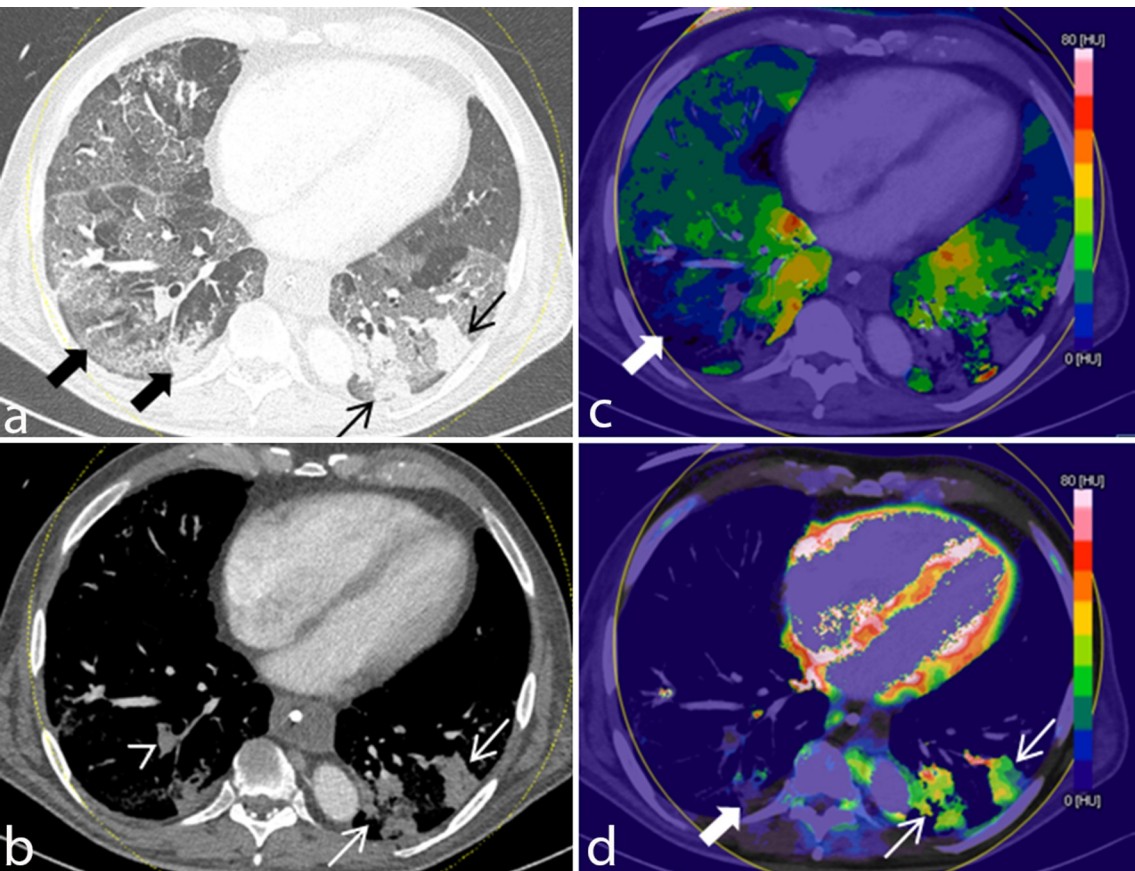

**Fig 4. Iodine uptake in normally aerated lung and opacifications with and without upstream pulmonary embolism.** Four reconstructions of the identical slice position, a. MPR in lung window, b. MPR in soft tissue window, c. colour-coded PBV MIP with thresholds for tissue recognition set to minimum -960 HU and maximum -400 HU for visualisation of iodine density in normally aerated lung and GGO and d. colour-coded PBV MIP with thresholds for tissue recognition set to minimum -100 HU and maximum +150 HU for visualization of iodine density in consolidation. Parameter settings were separated for this illustration only in order to enable a clear depiction of perfusion differences by optimizing the range of the look-up table to tissue density. Extensive diffuse GGO is present. Several consolidations are seen in the posterior segments of both lower lobes. Subsegmental pulmonary embolism is identified in the right lower lobe (arrowhead). Iodine density of lung tissue downstream of PE (bold arrows) is decreased while iodine uptake in GGO and in particular in consolidations (thin arrows) without upstream PE is increased compared to normally aerated lung (see colour- coding scale).

## Discussion

Since December 2019 the world suffers from the new Coronavirus SARS-CoV 2 induced air-way disease. Despite extensive scientific work-up the pathophysiology of lung injury caused by COVID-19 pneumonia is subject of ongoing controversy [1–4]. The discussion is mainly driven by the observation of an unusual ventilation pattern in terms of a combination of preserved lung compliance with severely impaired gas exchange. It was hypothesized that SARS-CoV 2 infection leads to significant vascular obstruction due to pulmonary embolism and / or in-situ thrombosis [28]. The resulting dead-space ventilation could explain the observed hypoxaemia. Meanwhile this hypothesis has been supported by autopsy results in several case series showing extensive microthrombi [13, 14]. However, an alternative explana-tion for hypoxaemia is significant shunting. Vessel enlargement is frequently observed within the opacified lung parenchyma of patients with COVID-19 pneumonia [16–18]. This feature seems to be unique for SARS-CoV 2 and has not been shown for other types of pneumonia. Several study groups evaluated lung perfusion based on the analysis of DECT data [22–26, 29].

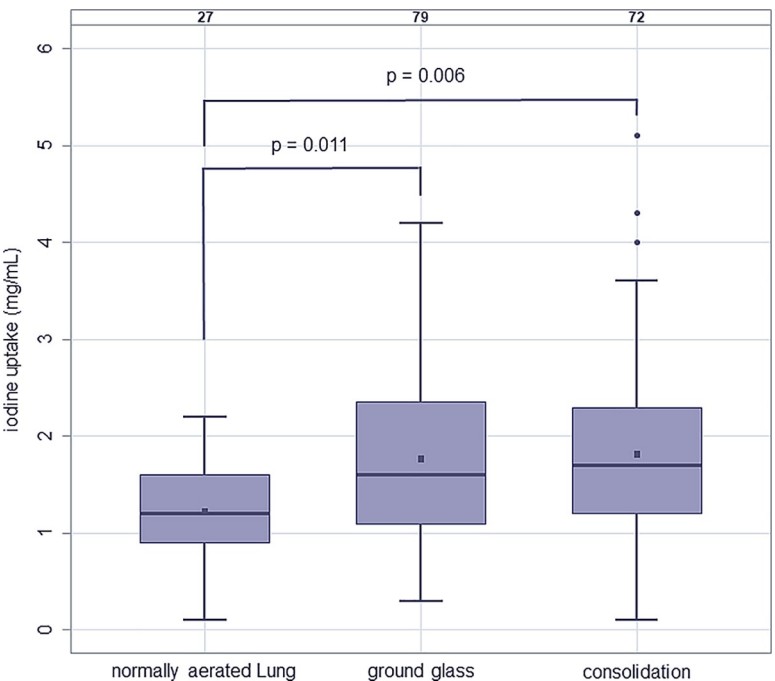

**Fig 5. Iodine uptake (mg/mL) of normally aerated lung (estimated mean: 1.22; 95%-CI: 0.95–1.49), areas of ground-glass opacification (estimated mean: 1.77; 95%-CI: 1.52–2.02), and consolidation (estimated mean: 1.82; 95%-CI: 1.56–2.08) without upstream pulmonary embolism.**

The results are inconsistent probably due to different study setups and even more so inhomogeneous study populations and disease stages. However, meanwhile it is well known that COVID-19 presents with a complex pathophysiological dynamic traversing the stages of early infection, pulmonary manifestation, thrombo-inflammation and hyperinflammation [30–37]. Thus it seems not to be surprising that pulmonary blood flow is subject to fluctuations during course of disease.

We systematically evaluated a cohort of 27 patients with RT-PCR proven COVID-19 pneumonia being treated in a tertiary care university hospital. All except one patient required intensive care, 89% were mechanically ventilated. Thus our cohort represents the subset of advanced and severe disease stage. We focused on the subgroup of severely diseased patients in order a) to establish a homogeneous study group in terms of stage of lung injury and b) to investigate a study group at risk with a strong need for further pathophysiological insight. DECT demonstrated a wide range of perfusion characteristics of pulmonary opacifications (both GGO and consolidation) with some being hypo- and others being hyperperfused. Increased iodine uptake was associated with an increase of intralesional vessel diameter. This feature has been described as typical for COVID-19 pneumonia [16–18]. Expectedly, opacifications downstream of pulmonary embolism demonstrated decreased iodine uptake.

Our results raise new aspects regarding the nature of lung injury caused by SARS-CoV-2 in the advanced stage. On one hand, we saw opacifications that were hypoperfused as compared to normally aerated lung. This finding is in line with the investigations of Lang et al. and Si-Mohamed et al. and supports the hypothesis of dead-space ventilation as one cause of impaired gas exchange [22, 25, 26].

On the other hand, we were able to demonstrate that a considerable fraction of opacifications is hyperperfused despite being diminished or non-ventilated. This finding suggests the presence of a functional intrapulmonary right-to-left shunt. Estimation of the amount of

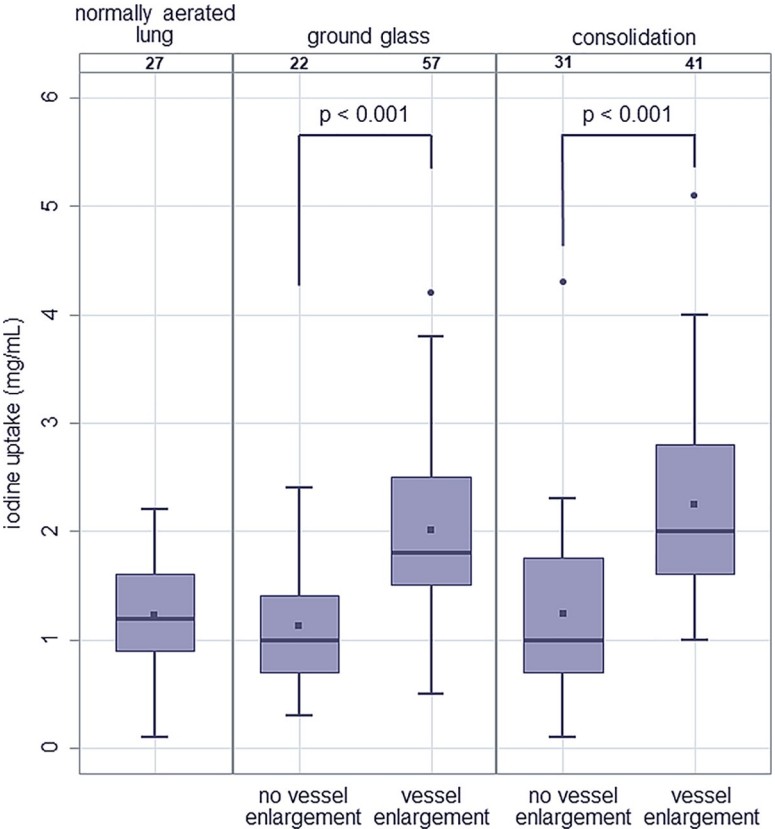

**Fig 6. Iodine uptake (mg/mL) of normally aerated lung (estimated mean: 1.22; 95%-CI: 0.95–1.49), areas of ground-glass opacification without and with vessel enlargement (estimated mean: 1.15; 95%-CI: 0.79–1.15; and 2.01; 95%-CI: 1.74–2.28, respectively), and areas of consolidation without and with vessel enlargement (estimated mean: 1.21; 95%-CI: 0.90–1.53; and 2.28; 95%-CI: 1.98–2.57, respectively).**

hyperperfused lung tissue on CTs performed without dual-energy technique might be possible by evaluation of the prevalence of vessel enlargement the presence of which correlated with hyperperfusion. An in-depth pathophysiological discussion is beyond the scope of the article. However, a direct effect of SARS-CoV-2 on the vascular autoregulation has been discussed and attributed to the virus' affinity to the ACE-2 receptor [38]. Thus, hyperaemia is possibly mediated by vasodilation or failed hypoxic vasoconstriction [2]. Also, neo-angiogenesis was described in COVID-19 pneumonia in a recently published autopsy series [13]. Our findings might help to tailor the therapeutic management to the individual patient's situation. Patients with predominant hypoperfusion / deadspace ventilation and/or pulmonary embolism will likely benefit from inhalative vasodilators (e.g. NO or prostacycline) which on the other hand might be deleterious for patients with mainly hyperperfused opacifications and vessel enlargement.

Limitations of our study are as follows: The sample size was rather small and the study design was retrospective. Thus our results have to be confirmed in future studies. More importantly, the conclusions drawn for clinical management have to be carefully evaluated. Selection of measurement areas for perfusion analysis was subjective and the ROIs represented a sample of the entire volume of opacifications. However, placing of ROIs were guided by the colour-coded iodine map enabling registration of the entire range PBV. Artificial intelligence-enhanced computing algorithms could enable the registration of iodine uptake over the entire

extent of opacifications. Such a software, however, was not available at our institution. The DECTs were acquired with different protocols regarding iodine delivery rate. This was due the fact that DECTs from clinical routine were analyzed. However, results were presented and analyzed with regard to the relative difference of iodine uptake in opacifications compared to normally aerated lung. This way an intraindividual normalization was established.

In summary, we were able to demonstrate that opacifications in patients with severe COVID-19 pneumonia can be both hypoperfused and hyperperfused. Indeed, the mean iodine uptake within pulmonary opacifications in our cohort was higher than that in normally aerated lung. Thus, impaired gas exchange might be caused by a combination of dead-space ventilation and functional intrapulmonary right-to-left shunt. Of note, intralesional vessel enlargement correlated with iodine uptake. Further research including correlation of perfusion imaging with ventilation parameters, minimal-invasive shunt assessment via right heart catheterization and histopathology is mandatory for a better understanding of the pathophysiology of COVID-19 pneumonia.

## Conclusion

Assessment of lung perfusion with DECT in advanced COVID-19 pneumonia revealed opacifications to be both hypo- and hyperperfused. Thus, a combination of dead-space ventilation and functional intrapulmonary right-to-left shunt might contribute to impairment of gas exchange. Applying DECT to determine which pathophysiology is predominant might help to tailor therapy to the individual patient´s needs.

## Supporting information

**S1 Data.**
(XLSX)

## Author Contributions

**Conceptualization:** Florian Poschenrieder, Matthias Lubnow, Okka W. Hamer.

**Data curation:** Florian Poschenrieder, Stefanie Meiler, Florian Zeman, Jan Schaible.

**Formal analysis:** Florian Zeman.

**Investigation:** Florian Poschenrieder, Stefanie Meiler, Janine Rennert, Jan Schaible.

**Methodology:** Florian Poschenrieder, Florian Zeman, Okka W. Hamer.

**Project administration:** Florian Poschenrieder, Stefanie Meiler.

**Resources:** Christian Stroszczynski, Michael Pfeifer.

**Software:** Florian Zeman, Gregor Scharf.

**Supervision:** Florian Poschenrieder, Okka W. Hamer.

**Validation:** Florian Poschenrieder, Janine Rennert, Okka W. Hamer.

**Visualization:** Florian Poschenrieder, Florian Zeman, Okka W. Hamer.

**Writing – original draft:** Florian Poschenrieder, Okka W. Hamer.

**Writing – review & editing:** Matthias Lubnow, Okka W. Hamer.

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
