## [Decision Letter · Decision Letter 0]

3 Feb 2021

PONE-D-20-38572

Advanced COVID-19 Pneumonia: Perfusion Analysis in Correlation with Pulmonary Embolism and Vessel Enlargement using Dual-Energy CT Data

PLOS ONE

Dear Dr. Poschenrieder,

Thank you for submitting your manuscript to PLOS ONE. After careful consideration, we feel that it has merit but does not fully meet PLOS ONE’s publication criteria as it currently stands. Therefore, we invite you to submit a revised version of the manuscript that addresses the points raised during the review process.

As you will recognize from the comments of the reviewers both raised points of critique, especially regarding methodology and discussion of findings.

Please submit your revised manuscript within 2 months. If you will need more time than this to complete your revisions, please reply to this message or contact the journal office at plosone@plos.org. Please include the following items when submitting your revised manuscript:

We look forward to receiving your revised manuscript.

Kind regards,

Rudolf Kirchmair

Academic Editor

PLOS ONE

Journal Requirements:

Reviewers' comments:

Reviewer's Responses to Questions

**Comments to the Author**

1. Is the manuscript technically sound, and do the data support the conclusions?

Reviewer #1: Yes

Reviewer #2: Partly

2. Has the statistical analysis been performed appropriately and rigorously? 

Reviewer #1: Yes

Reviewer #2: Yes

3. Have the authors made all data underlying the findings in their manuscript fully available?

Reviewer #1: Yes

Reviewer #2: No

4. Is the manuscript presented in an intelligible fashion and written in standard English?

Reviewer #1: Yes

Reviewer #2: Yes

5. Review Comments to the Author

Reviewer #1: This is a retrospective study evaluating DECT derived iodine uptake in 27 patients with severe COVID-19. The results show a predominance of hyperperfusion in opacified areas which correlates with vessel enlargement.

Major remarks:

Material and Methods

L66: Patients. How did you clinically define “advanced” COVID-19 pneumonia and “severe” COVID-19 pneumonia? Please explain and keep only one of these surrogates.

L79: How many patients were excluded because of non-diagnostic CTs due to technical artifacts?

L84: Anticoagulation status. Please specify.

L92: CT protocol. Since most patients (89%) were mechanically ventilated, how did you manage to perform CT scans during end-inspiration? Please specify.

L130: Vessel enlargement. You did not separate arterial from venous vessels. Please denote.

L135: As shown in figures 3 and 4, the colour-coded PBV maps are very inhomogeneous. Did the 2-dimensional axial plane ROI contain the entire geographic area of GGO and consolidation? Please explain in more detail. In COVID-19, mixed areas of GGO with some band-like consolidations are frequently observed. Did you exclude these mixed areas? A 3D segmentation of GGO and consolidation would be technically possible and the parameters of these VOI could have been evaluated as well.

Results

L157: Mean interval from symptom onset to the date of the DECT ranged considerably from 7-62 days. What was the clinical indication for DECT? Critical worsening of symptoms? Suspicion of pulmonary embolism? Please give more details on the overall clinical situation and treatment (e.g. corticosteroids, oxgen therapy, ECMO, …). It would be also of great interest to compare early scans, e.g. within 14 days of symptoms from late scans.

L163: Please provide information on image quality. What was the HU obtained in the pulmonary trunk? Image quality in pulmonary CTAs may vary substantially. Motion artifacts are frequently observed in mechanically ventilated ICU patients, as well as in critical non-intubated patients who frequently have problems in holding their breath. Please give more details.

Reviewer #2: Poschenrieder and colleagues present an interesting retrospective study including a cohort of 27 patients with severe COVID-19 pneumonia, describing/evaluating lung perfusion on dual energy CT through the iodine uptake measure.

The main finding in this study was that opacifications in the subset of critically ill COVID-19 patients can be both hypoperfused and hyperperfused. Authors hypothesize that this critical finding is the combination of dead-space ventilation and intrapulmonary right-to-left shunt, respectively, and thus explaining the COVID-19 pattern with severely impaired gas exchange.

I have the following comments and suggestions:

Major points:

- Keywords are missing in the manuscript.

- Materials and Methods, Image Analysis: please explain deeper how many experienced thoracic radiologist read each imaging test, if it was blinded to patient status or not, and the kappa index between radiologists for the variables (vessel enlargement, opacifications, but mainly for pulmonary embolism).

- Discussion: It should be interesting that authors explain the potential clinical impact of their finding (i.e. anticoaugulation, anti-infllammatory).

- Discussion: although the discussion is quite well written, I consider that authors should reconsider important limitations of the methodology used (i.e. simple size; limitations of observational studies to infer causality, ...).

Minor points:

- Introduction, line 40: it is said that “aggressive anticoagulant is meanwhile accepted part of therapy”. Although it has been purposed by some authors, the reference used by authors (Bikdeli B, et al. J Am Coll Cardiol 2020) did not recommend this management, and this is a very controversial point with very low certainty (WHO document: Clinical Management of COVID-19: interim guidance).

We strongly recommend the authors to replace this reference by recent clinical trial that could support this sentence: Tacquard C, Mansour A, Godon A, et al, Impact of high dose prophylactic anticoagulation in critically ill patients with COVID-19 pneumonia, CHEST (2021), doi: https://doi.org/10.1016/j.chest.2021.01.017.

- Introduction, line 60: please correct “the reasons for the conflicting results”.

- Materials and Methods, line 125: please correct “consensus”.

- Discussion, line 227: please correct “… evaluated a cohort of ….”

- Discussion, line 250: please correct “… SARS-Cov-2 ….”

- Discussion, line 264: please correct “… that opacifications in the subset ….”

6. PLOS authors have the option to publish the peer review history of their article (what does this mean?). If published, this will include your full peer review and any attached files.

Reviewer #1: No

Reviewer #2: No

---

## [Author Response · Author response to Decision Letter 0]

7 Apr 2021

Hereafter we are pleased to respond to each point raised by the academic editor and the reviewers:

Journal Requirements

1. We thoroughly reviewed our manuscript text including the manuscript title and adjusted it to PLOS ONE style requirements.

2. We removed the redundant ethics statement from the statement section at the end oft he mansucript. As you suggested, the ethics statement now appears only in the methods section of our manuscript.

Reply to review comments

@Reviewer #1:

Material and Methods:

L66: Patients. How did you clinically define “advanced” COVID-19 pneumonia and “severe” COVID-19 pneumonia? Please explain and keep only one of these surrogates.

Response: Thank you very much for this comment. We used the specifications „advanced“ and „severe“ interchangeable. The definition of severe pneumonia is outlined in the MM section. We now replaced „advanced“ by „severe“ in the whole manuscript text and title for the sake of consistency. 

L79: How many patients were excluded because of non-diagnostic CTs due to technical artifacts?

Response: During the survey period from April 1st and May 15th 2020, 29 patients with COVID-19 pneumonia underwent dual-energy CT and thus met the inclusion criteria. Image quality of all examinations was rated before analysis. Two patients had to be excluded due to technical artifacts („zebra artifacts“), so that finally 27 patients could be included for pulmonary perfusion analysis. We added this information as part of a more precise description of patient selection and image quality assessment.

L84: Anticoagulation status. Please specify.

Response: Reviewing our medical files, we now described in detail in the MM section how anticoagulation status was assessed. We also added the respective data to the results section of our manuscript.

L92: CT protocol. Since most patients (89%) were mechanically ventilated, how did you manage to perform CT scans during end-inspiration? Please specify.

Response: Thank your very much for this important remark. All dual-energy CTs were performed performing an inspiratory-hold maneuver. We added this information.

L130: Vessel enlargement. You did not separate arterial from venous vessels. Please denote.

Response: We are grateful for this important comment which gives us the opportunity to clarify. To the best of our knowledge differentiation between arterial and venous vessel enlargement was not done in literature so far. Although this would be an interesting point, the value for clarifying pathophysiology would probably be limited. Vessel enlargement can occur both in pulmonary embolism / in-situ thrombosis and hyperaemia. We believe that pulmonary blood volume is a better parameter to separate these conditions. This was one of the main objectives of our study. According to our results vessel enlargement reflects rather hyperaemia of affected lung areas than thrombus formation /embolism.

L135: As shown in figures 3 and 4, the colour-coded PBV maps are very inhomogeneous. Did the 2-dimensional axial plane ROI contain the entire geographic area of GGO and consolidation? Please explain in more detail. In COVID-19, mixed areas of GGO with some band-like consolidations are frequently observed. Did you exclude these mixed areas? A 3D segmentation of GGO and consolidation would be technically possible and the parameters of these VOI could have been evaluated as well.

Response: Thank you for this comment. The colour-coded PBV maps were displayed as axial maximum intensity projections at 5 millimeters reconstruction slice thickness. The ROIs for consolidation and GGO were drawn as large as possible. We intentionally exlcuded mixed areas in order to collect data purely representing consolidation and GGO, respectively. Unfortunately, the software did not provide the technical requirements for VOI-based three-dimensional analysis of dual-energy CT data.

Results

L157: Mean interval from symptom onset to the date of the DECT ranged considerably from 7-62 days. What was the clinical indication for DECT? Critical worsening of symptoms? Suspicion of pulmonary embolism? 

Response: Thank you very much for this comment. The clinical indication for DECT was critical worsening of symptoms in 25 out of 27 patients (92.6 %) and suspicion of pulmonary embolism in 5 out of 27 patients (18.5 %). We added this information to the results. 

Please give more details on the overall clinical situation and treatment (e.g. corticosteroids, oxgen therapy, ECMO, …).

Response: According to the reviewer´s point of criticism we specified clinical data and now report treatment data in detail. 

It would be also of great interest to compare early scans, e.g. within 14 days of symptoms from late scans. 

Response: Thank you very much for addressing this interesting issue. The vast majority of our patients underwent dual-energy CT in a late stage of disease, with only 5 patients out of 27 (18.5 %) being scanned within 14 days after symptom onset, versus 22 out of 27 patients (81.5 %) being examined later than 14 days after symptom onset. Thus, a comparison of these two groups unfortunately was not possible.

L163: Please provide information on image quality. What was the HU obtained in the pulmonary trunk? Image quality in pulmonary CTAs may vary substantially. Motion artifacts are frequently observed in mechanically ventilated ICU patients, as well as in critical non-intubated patients who frequently have problems in holding their breath. Please give more details.

Response: Thank you very much for bringing up this important point. As mentioned above 2 out of 29 dual-energy CTs had to be excluded from our study due to heavy technical artifacts („zebra artifacts“) which obviated evaluation of pulmonary perfusion pattern. We now amended and re-structurized the materials and methods section of our manuscript. As part of this, we systematically evaluated overall image quality considering motion and beam hardening artifacts and contrast-noise ratio. Overall image quality was subjectively assessed by two readers in consensus, one board-certified general radiologist with 12 years of experience (JR) and one subspeciality thoracic radiologist with 15 years of experience (FP), using a four-point scale. The results of this additional evaluation were added to the results section of our manuscript.

@Reviewer #2:

Major points:

- Keywords are missing in the manuscript.

Response: We added the keywords.

- Materials and Methods, Image Analysis: please explain deeper how many experienced thoracic radiologist read each imaging test, if it was blinded to patient status or not, and the kappa index between radiologists for the variables (vessel enlargement, opacifications, but mainly for pulmonary embolism).

Response: Thank you very much for your this important comment which gives us the opportunity to clarify. The image analysis was done by two readers in consensus, a board-certified general radiologist with 12 years of experience (JR) and a subspeciality thoracic radiologist with 15 years of experience (FP). Both readers were blinded to patient status. We amended the materials and methods section.

- Discussion: It should be interesting that authors explain the potential clinical impact of their finding (i.e. anticoagulation, anti-inflammatory).

Response: Thank you very much for addressing the potential clinical impact of our findings. We discussed this point extensively with our clinical colleagues who had encouraged us to do this study in the first place. Our findings might help to tailor the therapeutic management to the individual patient´s situation. Patients with predominant hypoperfusion /deadspace ventilation and /or pulmonary embolism will likely benefit from inhalative vasodilators (e.g. NO or prostacycline) which on the other hand might be deleterious for patients with mainly hyperperfused opacifications and vessel enlargement. We amended th discussion by these considerations.

- Discussion: although the discussion is quite well written, I consider that authors should reconsider important limitations of the methodology used (i.e. simple size; limitations of observational studies to infer causality, ...).

Response: We now amended the limitations paragraph in the discussion according to your helpful suggestions. 

Minor points:

- Introduction, line 40: it is said that “aggressive anticoagulant is meanwhile accepted part of therapy”. Although it has been purposed by some authors, the reference used by authors (Bikdeli B, et al. J Am Coll Cardiol 2020) did not recommend this management, and this is a very controversial point with very low certainty (WHO document: Clinical Management of COVID-19: interim guidance).

We strongly recommend the authors to replace this reference by recent clinical trial that could support this sentence: Tacquard C, Mansour A, Godon A, et al, Impact of high dose prophylactic anticoagulation in critically ill patients with COVID-19 pneumonia, CHEST (2021), doi: https://doi.org/10.1016/j.chest.2021.01.017.

Response: Thank you very much for correcting us and providing new data addressing this critical point. We replaced the respective reference by the suggested one.

- Introduction, line 60: please correct “the reasons for the conflicting results”.

Response: Thank you very much for your alertness. We apologize for this oversight and corrected it. 

- Materials and Methods, line 125: please correct “consensus”.

Response: Thank you very much for your alertness. We apologize for this oversight and corrected it. 

- Discussion, line 227: please correct “… evaluated a cohort of ….”

Response: Thank you very much for your alertness. We apologize for this oversight and corrected it. 

- Discussion, line 250: please correct “… SARS-Cov-2 ….”

Response: Thank you very much for your alertness. We apologize for this oversight and corrected it. 

- Discussion, line 264: please correct “… that opacifications in the subset ….”

Response: Thank you very much for your alertness. We apologize for this oversight and corrected it.

---

## [Decision Letter · Decision Letter 1]

17 May 2021

Severe COVID-19 pneumonia: perfusion analysis in correlation with pulmonary embolism and vessel enlargement using dual-energy CT data

PONE-D-20-38572R1

Dear Dr. Poschenrieder,

We’re pleased to inform you that your manuscript has been judged scientifically suitable for publication and will be formally accepted for publication once it meets all outstanding technical requirements.

Kind regards,

Rudolf Kirchmair

Academic Editor

PLOS ONE

Additional Editor Comments (optional):

Reviewers' comments:

Reviewer's Responses to Questions

**Comments to the Author**

1. If the authors have adequately addressed your comments raised in a previous round of review and you feel that this manuscript is now acceptable for publication, you may indicate that here to bypass the “Comments to the Author” section, enter your conflict of interest statement in the “Confidential to Editor” section, and submit your "Accept" recommendation.

Reviewer #1: All comments have been addressed

Reviewer #2: All comments have been addressed

2. Is the manuscript technically sound, and do the data support the conclusions?

Reviewer #1: Yes

Reviewer #2: Yes

3. Has the statistical analysis been performed appropriately and rigorously? 

Reviewer #1: Yes

Reviewer #2: Yes

4. Have the authors made all data underlying the findings in their manuscript fully available?

Reviewer #1: Yes

Reviewer #2: Yes

5. Is the manuscript presented in an intelligible fashion and written in standard English?

Reviewer #1: Yes

Reviewer #2: Yes

6. Review Comments to the Author

Reviewer #1: (No Response)

Reviewer #2: The authors have provided adequate responses to most of my concerns. I congratulate the authors for it.

7. PLOS authors have the option to publish the peer review history of their article (what does this mean?). If published, this will include your full peer review and any attached files.

Reviewer #1: **Yes: **Gerlig Widmann

Reviewer #2: **Yes: **Alberto García-Ortega

---

## [Editor Report · Acceptance letter]

31 May 2021

PONE-D-20-38572R1 

Severe COVID-19 pneumonia: perfusion analysis in correlation with pulmonary embolism and vessel enlargement using dual-energy CT data 

Dear Dr. Poschenrieder:

I'm pleased to inform you that your manuscript has been deemed suitable for publication in PLOS ONE. Congratulations! Your manuscript is now with our production department. 

Kind regards, 

on behalf of

Prof Rudolf Kirchmair 

Academic Editor

PLOS ONE